# Genetic and Phenotypic Spectrum of *KMT2D* Variants in Taiwanese Case Series of Kabuki Syndrome

**DOI:** 10.3390/diagnostics14161815

**Published:** 2024-08-20

**Authors:** Chung-Lin Lee, Chih-Kuang Chuang, Ming-Ren Chen, Ju-Li Lin, Huei-Ching Chiu, Ya-Hui Chang, Yuan-Rong Tu, Yun-Ting Lo, Hsiang-Yu Lin, Shuan-Pei Lin

**Affiliations:** 1Department of Pediatrics, MacKay Memorial Hospital, Taipei 10449, Taiwan; clampcage@gmail.com (C.-L.L.); mingren44@gmail.com (M.-R.C.); g880a01@mmh.org.tw (H.-C.C.); wish1001026@gmail.com (Y.-H.C.); 2Institute of Clinical Medicine, National Yang-Ming Chiao-Tung University, Taipei 112304, Taiwan; 3Department of Rare Disease Center, MacKay Memorial Hospital, Taipei 10449, Taiwan; andy11tw.e347@mmh.org.tw; 4Department of Medicine, Mackay Medical College, New Taipei City 25245, Taiwan; 5Department of Nursing, Mackay Junior College of Medicine, Nursing and Management, Taipei 112021, Taiwan; 6Division of Genetics and Metabolism, Department of Medical Research, MacKay Memorial Hospital, Taipei 10449, Taiwan; mmhcck@gmail.com (C.-K.C.); likemaruko@hotmail.com (Y.-R.T.); 7College of Medicine, Fu-Jen Catholic University, New Taipei City 24205, Taiwan; 8Division of Endocrine & Medical Genetics, Department of Pediatrics, Chang Gung Children’s Medical Center, Chang Gung Memorial Hospital, Taoyuan 33378, Taiwan; jllin001@gmail.com; 9Department of Medical Research, China Medical University Hospital, China Medical University, Taichung 40402, Taiwan; 10Department of Infant and Child Care, National Taipei University of Nursing and Health Sciences, Taipei 11219, Taiwan

**Keywords:** Kabuki syndrome, *KMT2D*, phenotypic heterogeneity, Taiwan

## Abstract

Kabuki syndrome (KS) is a rare genetic disorder characterized by distinct facial features, intellectual disability, and multiple congenital anomalies. We conducted a comprehensive analysis of the genetic and phenotypic spectrum of KS in a Taiwanese patient group of 23 patients. *KMT2D* variants were found in 22 individuals, with missense (26.1%), nonsense (21.7%), and frameshift (17.4%) variants being the most prevalent. One patient had a *KMT2D* variant of uncertain significance. The most common clinical characteristics included distinct facial features (100%), intellectual disability (100%), developmental delay (95.7%), speech delay (78.3%), hypotonia (69.6%), congenital heart abnormalities (69.6%), and recurrent infections (65.2%). Other abnormalities included hearing loss (39.1%), seizures (26.1%), cleft palate (26.1%), and renal anomalies (21.7%). This study broadens the mutational and phenotypic spectrum of KS in the Taiwanese population, highlighting the importance of comprehensive genetic testing and multidisciplinary clinical evaluations for diagnosis and treatment.

## 1. Introduction

Kabuki syndrome (KS) is a rare genetic disorder characterized by distinct facial features, intellectual disability, and various congenital anomalies. The cardinal facial features include long palpebral fissures with eversion of the lower eyelid’s lateral third, arched and wide eyebrows with sparseness in the lateral one-third, short columella with depressed nasal tip, and large, prominent ears [1,2]. Other common findings include persistent fetal finger pads, mild-to-moderate intellectual disability, postnatal growth deficiency, congenital heart defects, and genitourinary anomalies [1,2,3]. Recent studies have identified less common features such as immunological defects, endocrine abnormalities, and an increased risk of certain malignancies, further expanding our understanding of KS’s clinical presentation [4,5,6]. This broadening phenotypic spectrum highlights the complexity of KS and the need for comprehensive, multidisciplinary care for affected individuals.

KS is expected to affect one in every 32,000 births in Japan [4]. KS is caused by heterozygous pathogenic mutations in *KMT2D* (MLL2) or *KDM6A*, which encode histone-modifying enzymes [5,6]. *KMT2D* pathogenic variants are found in 56–76% of patients, whereas *KDM6A* pathogenic variants are found in 5–8% of patients [1,7]. Most variants in both genes are de novo truncating mutations that cause haploinsufficiency [8,9]. Recent research has broadened the genetic landscape of KS, identifying pathogenic variants in genes other than the well-established *KMT2D* and *KDM6A*. Notably, a few patients have been identified to have variants in *RAP1A*, *RAP1B*, and *HNRNPK*, indicating that these genes may contribute to the molecular etiology of KS in some cases [10,11,12]. However, the precise role and prevalence of these newly identified genes in the pathogenesis of KS have to be determined.

Establishing a molecular diagnosis in suspected KS patients is critical for accurate genetic counseling and proactive care. Present diagnostic approaches include targeted sequencing of *KMT2D* and *KDM6A*, followed by deletion/duplication analysis if the initial testing is negative [1]. Recently, a clinical scoring system and consensus diagnostic criteria were developed to help diagnose KS [2,13]. This scoring system has significantly improved early diagnosis and standardized clinical assessment of KS patients, facilitating more timely and appropriate interventions. However, the phenotypic spectrum of KS continues to broaden, and genotype–phenotype relationships are not fully elucidated.

This study includes a patient group of 23 Taiwanese patients with molecularly confirmed KS. Twenty-two had pathogenic or likely pathogenic variants in *KMT2D*, and one patient had a variant of uncertain significance in the same gene. Our primary objectives are to characterize the mutational landscape and clinical symptoms of KS in the Taiwanese population and investigate potential genotype–phenotype relationships. By understanding the genetic and phenotypic variability of KS in this patient group, we aim to contribute to the growing body of knowledge about the heterogeneity of this rare disorder. Our findings may have implications for improving the diagnosis, treatment, and genetic counseling of KS patients in Taiwan and elsewhere.

## 2. Materials and Methods

### 2.1. Patient Group and Data Collection

We conducted a retrospective analysis of 23 Taiwanese patients with molecularly confirmed KS assessed at MacKay Memorial Hospital between 2012 and 2023. Our study included 23 patients (12 males, 11 females) with an age range of 2–25 years at the time of genetic diagnosis. All patients were of Taiwanese descent. The diagnosis was made by experienced medical geneticists based on the presence of characteristic clinical features that met the diagnostic criteria reported in the literature [3,4,5]. All patients underwent comprehensive clinical evaluations, which included thorough physical examinations. Relevant clinical and laboratory data were collected and evaluated, including cardiac assessments (echocardiography).

### 2.2. Molecular Analysis

Peripheral blood leukocytes were collected from patients, and genomic DNA was isolated using standard protocols. Targeted resequencing of the *KMT2D* was performed using the custom-designed Truseq Custom Amplicon panel and the MiSeq sequencing platform (Illumina, San Diego, CA, USA). The panel had 33,255 base pairs and 135 amplicons, each spanning 500 base pairs, for 99% coverage of the target areas. Variant calling and visualization were performed using Illumina Variant Studio and Integrative Genome Viewer. Sanger sequencing was used to validate all identified variants. The pathogenicity of novel missense variants was determined using an in silico prediction tool (Alamut Software). When possible, parental testing was performed to determine whether the variants were de novo or inherited. Pathogenic variants were defined as those that resulted in protein truncation, altered splicing, or missense changes that were proven de novo in at least one patient. In cases where parental samples were unavailable, missense variants were considered pathogenic if they appeared in several patients or met the ACMG pathogenicity criteria. Variants of uncertain significance and those inherited from an unaffected parent were excluded from the study. Because all patients in our patient group were identified before 2013, genetic testing was limited to the *KMT2D* and did not include other genes subsequently associated with KS.

Bioinformatic analysis was performed using the Illumina BaseSpace Sequence Hub. Sequence reads were aligned to the human reference genome (GRCh38) using the Burrows–Wheeler aligner (BWA). Variant calling was performed using the Genome Analysis Toolkit (GATK). Quality control metrics included a minimum read depth of 20× and a mapping quality score of ≥30. Population frequencies of genetic variants were evaluated using gnomAD, ExAC, and 1000 Genomes databases. All variants were reported using the NM_003482.4 transcript of the *KMT2D*.

### 2.3. Study Limitations

Although the study period extended to 2023, genetic testing was limited to the *KMT2D* because all patients were initially identified before 2013, before the broader genetic landscape of KS was fully understood. This limitation may impact our study’s ability to fully capture the complete genetic heterogeneity of Kabuki syndrome, particularly concerning genes identified in later studies. Despite this limitation, our focus on *KMT2D* variants provides valuable insights into the most common genetic cause of KS in the Taiwanese population.

### 2.4. Variant Classification

Variants were classified as ‘pathogenic’ when they met strong criteria such as being null variants (nonsense, frameshift, canonical splice sites) or previously established as causative. ‘Likely pathogenic’ classifications were assigned to variants with strong evidence of pathogenicity but lacking definitive proof, such as novel missense variants at highly conserved residues with supporting functional studies. This classification was performed according to the American College of Medical Genetics and Genomics (ACMG) guidelines for the interpretation of sequence variants [14].

## 3. Results

### 3.1. Molecular Findings

In this study, pathogenic or likely pathogenic variants in the *KMT2D* were identified in 22 out of 23 Taiwanese patients with Kabuki syndrome. The remaining patient carried a variant of uncertain significance (VUS) in *KMT2D*. The distribution of the 16 variant types is as follows: five nonsense (31.3%), three missense (18.7%), three frameshift (18.7%), three deletions (18.7%), one splicing (6.3%), and one insertion/deletion (6.3%). Among these variants, eight are classified as pathogenic (50.0%), six as likely pathogenic (37.5%), and two as VUS (12.5%) (Figure 1a–c, Table 1).

Among the 16 unique *KMT2D* variants identified in our patient group, five were novel and not previously reported in the literature or public databases. These novel variants include c.15953_56delTATT (p.Leu5318Serfs14); c.13040_13041delAG (p.Gln4347Argfs24); c.10741-7A>G, c.15802_15804delATT (p.Ile5268del); and c.16412+16delG. The pathogenicity of these novel variants was determined based on the ACMG guidelines, considering factors such as population frequency, in silico predictions, and segregation analysis when possible.

### 3.2. Clinical Features

The clinical characteristics of the 23 patients are shown in Table 2. All patients had distinct facial features of KS and intellectual capacity (100%). Other common features were developmental delay (95.7%), speech delay (78.3%), hypotonia (69.6%), congenital heart defects (69.6%), recurrent infections (65.2%), short stature (60.9%), and feeding difficulties (43.5%).

Among the 16 patients with congenital heart defects, the most common anomalies were atrial septal defect (37.5%), ventricular septal defect (18.8%) and aortic coarctation (18.8%). Other uncommon cardiac abnormalities included bicuspid aortic valve, persistent left superior vena cava, mitral valve prolapse, and patent ductus arteriosus.

Additional findings in our patient group were hearing impairment (39.1%), seizures (26.1%), cleft palate (26.1%), renal anomalies (21.7%), preauricular pits (17.4%), and ophthalmologic abnormalities (17.4%). Sacral dimples, cryptorchidism, hip dislocation, premature thelarche, gastrointestinal anomalies, and dental anomalies were also observed in isolated cases.

## 4. Discussion

The present study provides a comprehensive analysis of the genetic and clinical features of KS in a Taiwanese population. Among the 23 patients with molecularly confirmed KS, 22 individuals had pathogenic or likely pathogenic variants in the *KMT2D* gene, and one patient had a variant of uncertain significance. The prevalence of *KMT2D* variants in our patient group is consistent with previous reports, which show that *KMT2D* is the major causative gene for KS, accounting for 56–76% of cases [1,7].

In our study, the most common variant types were missense (26.1%), nonsense (21.7%), and frameshift (17.4%), which is comparable with the mutational spectrum reported in other populations [2,8,9]. We identified several novel variants, increasing the known KS mutational landscape in the Taiwanese population.

Clinically, our patients demonstrated a wide range of KS-specific characteristics. The most common findings included distinct facial features, intellectual disability, developmental delay, speech delay, hypotonia, congenital heart defects, and recurrent infections. In a large patient group study by Lehman et al. [15], intellectual disability was observed in 99% of KS patients, whereas hypotonia and feeding difficulties were evident in 98% and 84% of patients, respectively. These findings are consistent with the high prevalence of these features in our patient group. Congenital heart defects, particularly septal defects, and aortic coarctation were prevalent in our patient group, highlighting the importance of cardiac evaluation in KS patients.

Interestingly, we found a relatively high frequency of hearing impairment (39.1%) compared with previous studies [2,3]. This finding emphasizes the need for routine hearing assessments for Taiwanese KS patients. Other significant features in our patient group included seizures, cleft palate, renal anomalies, and preauricular pits, all of which have been previously described in KS [1,2,3].

When we compared our results with those from other Asian patient group, we found similarities and differences. So et al. [16] found that Chinese KS patients had a comparable prevalence of intellectual disability, hypotonia, and congenital heart defects. However, they found a higher prevalence of renal anomalies (47.6%) than our patient group (21.7%). In a Turkish study by Usluer et al. [2], the frequencies of facial features, intellectual disability, and congenital heart defects were comparable with our findings. Notably, they reported a higher rate of cleft palate (38.1%) than our patient group (26.1%).

Compared with our and Turkish patient groups, Priolo et al. [17] found that Italian KS patients had a lower prevalence of cleft palate (15.8%). However, our patients exhibited similar clinical symptoms to those in Priolo et al.’s study. Banka et al. [9] found a comparable prevalence of intellectual disability, hypotonia, and congenital heart defects in a large patient group of KS patients from the United Kingdom to our patient group. However, they found a higher prevalence of renal anomalies (38.2%) and joint laxity (83.8%) than our patient group [9]. These differences indicate the possibility that genetic and environmental factors influence the phenotypic expression of KS across different populations.

The variability in phenotypic expression found across different populations may be impacted by genetic and environmental factors. Environmental factors that might influence KS phenotypic expression in Taiwan could include dietary habits, such as high consumption of iodine-rich seafood, which might impact thyroid function, a system often affected in KS. Additionally, the subtropical climate and associated pathogen exposure patterns could potentially influence the frequency and severity of recurrent infections observed in KS patients. However, these hypotheses require further investigation to establish any definitive links. Future studies focusing on the interaction between genetic variants and environmental factors in KS patients from different populations could provide valuable insights into the mechanisms underlying the phenotypic variability observed in this syndrome. In addition to *KMT2D* and *KDM6A*, other genes such as *KDM1A*, *ASXL1*, and *EZH2* have been linked to Kabuki-like phenotypes in a small subset of patients [5,8]. These findings show that the genetic landscape of KS may be more complex than previously thought, and further studies are needed to elucidate the role of these genes in the pathogenesis of KS.

This study has several limitations. One limitation is the absence of genetic testing for other KS-associated genes, such as *KDM6A*, *RAP1A*, *RAP1B*, and *HNRNPK*. Our patients were diagnosed before 2013; hence, the genetic analysis was limited to the *KMT2D*. Future studies that use a broader genetic testing approach may provide a more comprehensive understanding of the genetic landscape of KS in the Taiwanese population.

## 5. Conclusions

Our findings are consistent with the recommendations of the worldwide consensus statement on diagnosing and managing KS, which emphasizes the importance of a multidisciplinary approach that includes geneticists, pediatricians, cardiologists, endocrinologists, and other specialists. Identifying novel variants broadens the mutational spectrum of KS, whereas detailed clinical characterization helps better understand the phenotypic variability associated with this disorder. These findings highlight the importance of comprehensive genetic testing and multidisciplinary clinical evaluations in diagnosing and treating KS patients. Further research is needed to investigate potential genotype–phenotype relationships and to understand the molecular mechanisms underlying the phenotypic heterogeneity observed in KS.

## Figures and Tables

**Figure 1 diagnostics-14-01815-f001:**
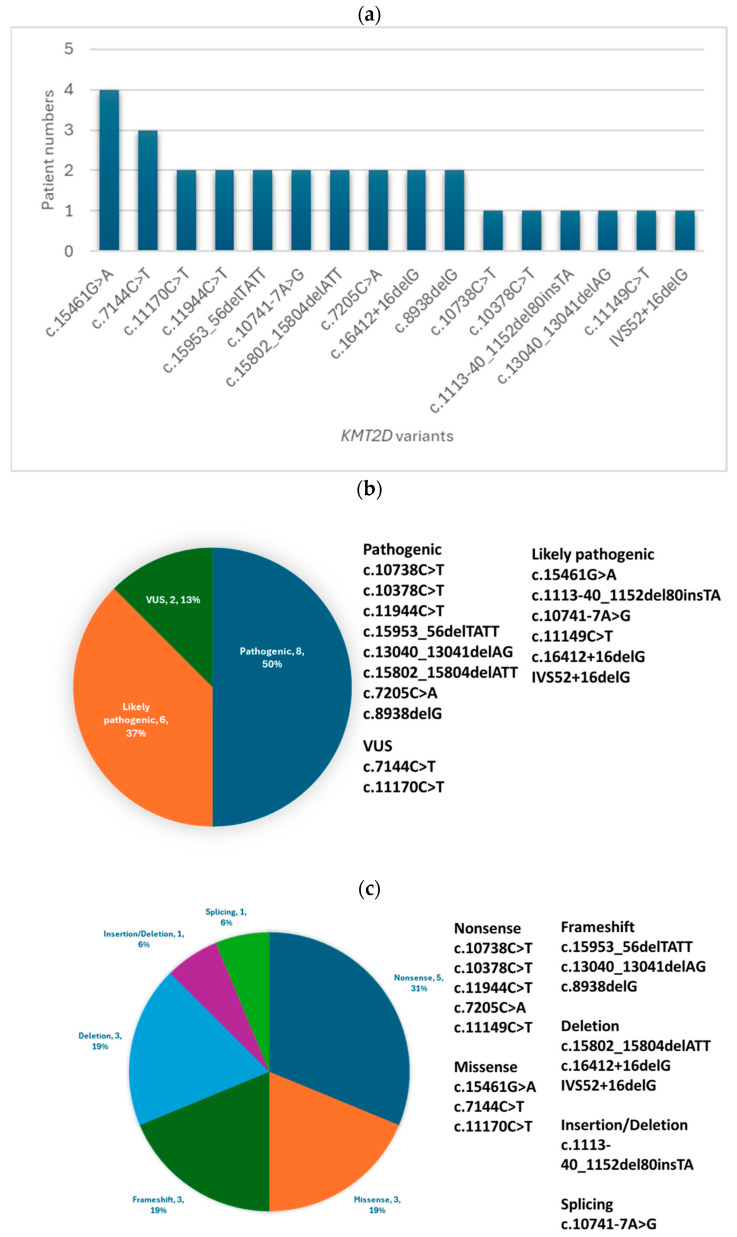
(**a**) The prevalence of *KMT2D* variants (16 variants) in Taiwanese patients with Kabuki syndrome. The bar graph illustrates the distribution of *KMT2D* variants in the study patient group. (**b**) Distribution of the pathogenicity classifications of *KMT2D* variants (16 variants) identified in the Taiwanese patient group. The pie chart illustrates the number of variants classified as pathogenic, likely pathogenic, and variants of uncertain significance (VUS). (**c**) Distribution of the different types of *KMT2D* variants (16 variants) identified in the Taiwanese patient group. The pie chart illustrates the proportion of missense, nonsense, frameshift, in-frame deletion, splicing, and gross deletion variants.

**Table 1 diagnostics-14-01815-t001:** Molecular characteristics of the Taiwanese patient group with *KMT2D* variants (NM_003482.4).

Patient	Sex	*KMT2D* cDNA Change	*KMT2D* Amino Acid Change	Pathogenicity	ACMG Classification	Inheritance
1	M	c.15461G>A	p.Arg5154Gln	Missense	Likely Pathogenic	De novo
2	M	c.10738C>T	p.Gln3580 *	Nonsense	Pathogenic	De novo
3	M	c.10378C>T	p.Gln3460 *	Nonsense	Pathogenic	De novo
4	M	c.11944C>T	p.Arg3982 *	Nonsense	Pathogenic	De novo
5	M	c.1113-40_1152del80insTA	-	Insertion/ Deletion	Likely Pathogenic	Unknown
6	F	c.15953_56delTATT	p.Leu5318Serfs *14	Frameshift	Pathogenic	De novo
7	F	c.13040_13041delAG	p.Gln4347Argfs *24	Frameshift	Pathogenic	De novo
		c.7144C>T	p.Pro2382Ser	Missense	Uncertain Significance	Unknown
8	M	c.10741-7A>G	-	Splicing	Likely Pathogenic	Unknown
		c.15802_15804delATT	p.Ile5268del	Deletion	Pathogenic	De novo
9	F	c.7205C>A	p.Ser2402 *	Nonsense	Pathogenic	De novo
10	F	c.15802_15804delATT	p.Ile5268del	Deletion	Pathogenic	De novo
11	M	c.7205C>A	p.Ser2402 *	Nonsense	Pathogenic	De novo
12	F	c.15953_56delTATT	p.Leu5318Serfs *14	Frameshift	Pathogenic	De novo
13	M	c.15461G>A	p.Arg5154Gln	Missense	Likely Pathogenic	De novo
14	M	c.11149C>T	p.Gln3717 *	Nonsense	Likely Pathogenic	De novo
15	F	c.7144C>T	p.Pro2382Ser	Missense	Uncertain Significance	Unknown
16	F	c.7144C>T	p.Pro2382Ser	Missense	Uncertain Significance	Unknown
		c.16412+16delG	-	Deletion	Likely Pathogenic	Unknown
17	F	c.15461G>A	p.Arg5154Gln	Missense	Likely Pathogenic	De novo
18	M	c.8938delG	p.Ala2980Profs *24	Frameshift	Pathogenic	De novo
19	M	c.11170C>T	p.Pro2382Ser	Missense	Uncertain Significance	Unknown
		c.16412+16delG	-	Deletion	Likely Pathogenic	Unknown
20	F	c.8938delG	p.Ala2980Profs *24	Frameshift	Pathogenic	De novo
21	F	c.15461G>A	p.Arg5154Gln	Missense	Likely Pathogenic	De novo
22	F	c.16412+16delG	-	Deletion	Likely Pathogenic	Unknown
23	M	c.10741-7A>G	-	Splicing	Likely Pathogenic	Unknown

Notes: All variants are reported using the NM_003482.4 transcript of the *KMT2D*. Asterisks (*) indicate a stop codon. Dashes (-) indicate that no amino acid change is associated with the corresponding nucleotide change. “Unknown” in the Inheritance column indicates that parental testing was not performed or was inconclusive.

**Table 2 diagnostics-14-01815-t002:** Clinical features of the 23 Taiwanese patients with Kabuki syndrome.

Clinical Feature	Number of Patients (Total = 23)	Percentage (%)
Distinct facial features	23	100
Intellectual disability	23	100
Developmental delay	22	95.7
Speech delay	18	78.3
Hypotonia	16	69.6
Congenital heart defects	16	69.6
ASD, ostium secundum	6/16	37.5
VSD, perimembranous subaortic	3/16	18.8
Aortic coarctation	3/16	18.8
Bicuspid aortic valve	2/16	12.5
Persistent left superior vena cava	2/16	12.5
Mitral valve prolapse	2/16	12.5
Mitral regurgitation	2/16	12.5
Patent ductus arteriosus	2/16	12.5
Mitral stenosis	1/16	6.3
Aberrant right subclavian artery	1/16	6.3
Vascular ring	1/16	6.3
Dilated aortic root	1/16	6.3
Interrupted aortic arch	1/16	6.3
Subaortic ridge	1/16	6.3
Left isomerism of the heart	1/16	6.3
Aortic dissection	1/16	6.3
Redundant mitral valve	1/16	6.3
Redundant tricuspid valve	1/16	6.3
Recurrent infections	15	65.2
Short stature	14	60.9
Feeding difficulties	10	43.5
Hearing impairment	9	39.1
Seizures	6	26.1
Cleft palate	6	26.1
Renal anomalies	5	21.7
Preauricular pits	4	17.4
Ophthalmologic abnormalities	4	17.4
Sacral dimple	2	8.7
Cryptorchidism	1	4.3
Hip dislocation	1	4.3
Premature thelarche	1	4.3
Gastrointestinal anomalies	1	4.3
Dental anomalies	1	4.3

## Data Availability

The original data and findings obtained during this study are fully reported in the manuscript. The corresponding authors can be contacted for additional data or information inquiries.

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
