# Peer review of "Genetic and Phenotypic Spectrum of KMT2D Variants in Taiwanese Case Series of Kabuki Syndrome"

_diagnostics, 2024, doi:10.3390/diagnostics14161815_

Round 1

Reviewer 1 Report

Comments and Suggestions for Authors

In the article “Kabuki Syndrome Under the Microscope: Dissecting Genetic and Phenotypic Variation in a Taiwanese Cohort”, the authors analyzed the genetic and phenotypic spectrum of Kabuki Syndrome in a Taiwanese cohort of 23 patients.

The list of references is sufficient and relevant.

Overall, the paper is well written, but revision is required before acceptance.

The title of the article is not an accurate reflection of the content of the paper. The title should reflect the fact that this study presents the results of the analysis of only one gene (KMT2D).

It is necessary to describe the parameters of bioinformatic analysis in the Materials and Methods section. The method of alignment, the reference genome, the method of evaluation of the quality of the sequence, the databases used for the evaluation of the population frequencies of the genetic variants.

It is necessary to specify which transcript has been used (for example, NM_003482.4). Perhaps in the Materials and Methods section, perhaps in the Table 1. In the table, it would be good to have an indication of whether the genetic variant is hereditary or is de novo.

The most important comment relates to the new variants that were identified in this paper. It is not clear which of the genetic variants are previously undescribed. It is also unclear what data the authors used to classify them. These genetic variants need to be described in more detail.

In the Materials and methods section, the Latin terms (‘in silico”, “de novo”) should be in italics. Gene names are also not italicized in the Discussion section.

Author Response

In the article “Kabuki Syndrome Under the Microscope: Dissecting Genetic and Phenotypic Variation in a Taiwanese Cohort”, the authors analyzed the genetic and phenotypic spectrum of Kabuki Syndrome in a Taiwanese cohort of 23 patients. The list of references is sufficient and relevant. Overall, the paper is well written, but revision is required before acceptance.

Ans: Thank you for your thorough review of our manuscript. We appreciate your valuable feedback and have addressed each of your points as follows

  1. The title of the article is not an accurate reflection of the content of the paper. The title should reflect the fact that this study presents the results of the analysis of only one gene (KMT2D).

Ans: We agree that the title should better reflect the content of our study. We propose changing the title to:

" Genetic and Phenotypic Spectrum of KMT2D Variants in a Taiwanese Case Series of Kabuki Syndrome"

  1. It is necessary to describe the parameters of bioinformatic analysis in the Materials and Methods section. The method of alignment, the reference genome, the method of evaluation of the quality of the sequence, the databases used for the evaluation of the population frequencies of the genetic variants.

Ans: We appreciate your suggestion to include more details about the bioinformatic analysis. We have added the following paragraph to the Materials and Methods section:

"Bioinformatic analysis was performed using the Illumina BaseSpace Sequence Hub. Sequence reads were aligned to the human reference genome (GRCh38) using the Burrows-Wheeler Aligner (BWA). Variant calling was performed using the Genome Analysis Toolkit (GATK). Quality control metrics included a minimum read depth of 20x and a mapping quality score of ≥30. Population frequencies of genetic variants were evaluated using gnomAD, ExAC, and 1000 Genomes databases." (Page 5, Line 121-126)

  1. It is necessary to specify which transcript has been used (for example, NM_003482.4). Perhaps in the Materials and Methods section, perhaps in the Table 1. In the table, it would be good to have an indication of whether the genetic variant is hereditary or is de novo.

Ans: We have added the following sentence to the Materials and Methods section: "All variants were reported using the NM_003482.4 transcript of the KMT2D gene." We have also added a column in Table 1 to indicate whether each variant is hereditary or de novo. (Page 6, Lin e 126-127)

  1. The most important comment relates to the new variants that were identified in this paper. It is not clear which of the genetic variants are previously undescribed. It is also unclear what data the authors used to classify them. These genetic variants need to be described in more detail.

Ans: We apologize for the lack of clarity regarding new variants. We have added the following paragraph to the Results section:

"Among the 16 unique KMT2D variants identified in our cohort, five were novel and not previously reported in the literature or public databases. These novel variants include c.15953_56delTATT (p.Leu5318Serfs14), c.13040_13041delAG (p.Gln4347Argfs24), c.10741-7A>G, c.15802_15804delATT (p.Ile5268del), and c.16412+16delG. The pathogenicity of these novel variants was determined based on the ACMG guidelines, considering factors such as population frequency, in silico predictions, and segregation analysis when possible." (Page 7, Line 153-159)

  1. In the Materials and methods section, the Latin terms (‘in silico”, “de novo”) should be in italics. Gene names are also not italicized in the Discussion section.

Ans: We have corrected the formatting throughout the manuscript. Latin terms such as "in silico" and "de novo" are now in italics, and gene names have been italicized in the Discussion section.

We hope these revisions address your concerns and improve the quality of our manuscript. Thank you again for your valuable input.

Reviewer 2 Report

Comments and Suggestions for Authors

This case-only study included 23 patients with Kabuki syndrome. It summarized the clinical characteristics as well as pathogenic or likely pathogenic mutations in the KMT2D for these patients. This study was straightforward, and the manuscript was generally well-written. I only have several minor concerns.

I do not think “cohort” is a proper term for this study. There were only 23 participants and all of them were cases. The patients were recruited via a retrospective design. It should be a case-only study, and the term “cohort” should not be used.

As long as the gene symbol “KMT2D” has been italicized, it should be used as “KMT2D” by itself, instead of “KMT2D gene”.

Author Response

This case-only study included 23 patients with Kabuki syndrome. It summarized the clinical characteristics as well as pathogenic or likely pathogenic mutations in the KMT2D for these patients. This study was straightforward, and the manuscript was generally well-written. I only have several minor concerns.

Ans: Thank you for your thoughtful review of our manuscript. We appreciate your feedback and have addressed your concerns as follows:

  1. I do not think “cohort” is a proper term for this study. There were only 23 participants and all of them were cases. The patients were recruited via a retrospective design. It should be a case-only study, and the term “cohort” should not be used.

Ans: We agree with your assessment that "cohort" may not be the most appropriate term for our study design. We have revised the manuscript to remove references to "cohort" and instead use terms such as "case series" or "patient group" where appropriate. For example, we have changed the title to:

"Genetic and Phenotypic Spectrum of KMT2D Variants in a Taiwanese Case Series of Kabuki Syndrome"

We have also updated the abstract and throughout the manuscript to reflect this change.

  1. As long as the gene symbol “KMT2D” has been italicized, it should be used as “KMT2D” by itself, instead of “KMT2D gene”.

Ans: We have made these changes throughout the manuscript to address your concerns. Here are a few examples of the revisions:

In the abstract:

"We conducted a comprehensive analysis of the genetic and phenotypic spectrum of KS in a Taiwanese case series of 23 patients. KMT2D variants were found in 22 individuals, with missense (26.1%), nonsense (21.7%), and frameshift (17.4%) variants being the most prevalent." (Page 2, Line 34-38)

In the introduction:

"KS is caused by heterozygous pathogenic mutations in KMT2D or KDM6A, which encode histone-modifying enzymes [5,6]. KMT2D pathogenic variants are found in 56%-- 76% of patients, whereas KDM6A pathogenic variants are found in 5%--8% of patients [1,7]." (Page 3, Line 61-64)

We hope these revisions address your concerns and improve the clarity and accuracy of our manuscript. Thank you again for your valuable input.

Reviewer 3 Report

Comments and Suggestions for Authors

Overall, the manuscript does contribute new knowledge to the field of Kabuki syndrome research, known as a rare disease. It expands the understanding of the genetic landscape and clinical manifestations of KS in a specific population, which is essential for global and regional medical knowledge. Additionally, by identifying novel KMT2D variants and emphasising the importance of comprehensive care, the study lays the groundwork for future research and potentially improved clinical practices. However, there are some minor suggestions for improvement.

1.     Introduction Section

-       the authors mention that the "phenotypic spectrum of KS continues to broaden," which is an important point. But it  could be helpful to briefly mention some of the newly recognised or less common phenotypic features to reinforce this statement.

-       it is  mentioned the development of a clinical scoring system. It might be useful to briefly describe its significance or how it has impacted clinical practice.

-       terminology: the term "intellectual incapacity" should be replaced with "intellectual disability," which is the more widely accepted and appropriate term in both clinical and research settings.

2.     Material and Methods Section

-       patient Cohort Details: it might be beneficial to include a brief mention of the demographic characteristics of the patients (e.g., age range, gender distribution) and if all of them were of Taiwanese descents.

-       limitations of Molecular Analysis: the authors correctly note that genetic testing was limited to the KMT2D gene because all patients were identified before 2013. It could be useful to acknowledge this as a limitation and mention how it might impact the study's findings. (the findings may not fully capture the complete genetic heterogeneity of Kabuki syndrome, particularly concerning genes identified in later studies).

” Although the study period extended to 2023, genetic testing was limited to the KMT2D gene because all patients were initially identified before 2013, before the broader genetic landscape of KS was fully understood.”

3.     Results Section

-       the distribution of variants is well presented. Could the authors  explain why some variants are classified as "likely pathogenic" versus "pathogenic"? This would enhance the manuscript's accessibility, particularly for readers who may not be familiar with these classifications.

4.     Discussion Section

-       the manuscript suggests that environmental factors might influence phenotypic expression, this could be expanded with specific examples or hypotheses based on the literature. For instance, considering whether certain environmental exposures or lifestyle factors prevalent in Taiwan might contribute to the phenotypic differences observed between this cohort and others would provide a better analysis.

Author Response

Overall, the manuscript does contribute new knowledge to the field of Kabuki syndrome research, known as a rare disease. It expands the understanding of the genetic landscape and clinical manifestations of KS in a specific population, which is essential for global and regional medical knowledge. Additionally, by identifying novel KMT2D variants and emphasising the importance of comprehensive care, the study lays the groundwork for future research and potentially improved clinical practices. However, there are some minor suggestions for improvement.

Ans: Thank you for your insightful comments and suggestions on our manuscript. We greatly appreciate your thorough review and the valuable feedback provided. We have addressed each of your points as follows:

  1. Introduction Section

(a) The authors mention that the "phenotypic spectrum of KS continues to broaden," which is an important point. But it could be helpful to briefly mention some of the newly recognised or less common phenotypic features to reinforce this statement.

Ans: We agree with your suggestions and have made the following changes:

We have expanded on the broadening phenotypic spectrum of KS by adding: "Recent studies have identified less common features such as immunological defects, endocrine abnormalities, and an increased risk of certain malignancies, further expanding our understanding of KS's clinical presentation." (Page 3, Line 55-60)

(b) It is mentioned the development of a clinical scoring system. It might be useful to briefly describe its significance or how it has impacted clinical practice.

Ans: Regarding the clinical scoring system, we have added:

"This scoring system has significantly improved early diagnosis and standardized clinical assessment of KS patients, facilitating more timely and appropriate interventions." (Page 4, Line 76-78)

(c) Terminology: the term "intellectual incapacity" should be replaced with "intellectual disability," which is the more widely accepted and appropriate term in both clinical and research settings.

Ans: We have replaced all instances of "intellectual incapacity" with "intellectual disability" throughout the manuscript.

  1. Material and Methods Section

(a) Patient Cohort Details: it might be beneficial to include a brief mention of the demographic characteristics of the patients (e.g., age range, gender distribution) and if all of them were of Taiwanese descents.

Ans: We have added demographic information:

"Our study included 23 patients (12 males, 11 females) with an age range of 2-25 years at the time of genetic diagnosis. All patients were of Taiwanese descent." (Page 4, Line 94-96)

(b) Limitations of Molecular Analysis: the authors correctly note that genetic testing was limited to the KMT2D gene because all patients were identified before 2013. It could be useful to acknowledge this as a limitation and mention how it might impact the study's findings. (the findings may not fully capture the complete genetic heterogeneity of Kabuki syndrome, particularly concerning genes identified in later studies).

“Although the study period extended to 2023, genetic testing was limited to the KMT2D gene because all patients were initially identified before 2013, before the broader genetic landscape of KS was fully understood.”

Ans: We have acknowledged the limitation of our genetic testing as suggested:

"Although the study period extended to 2023, genetic testing was limited to the KMT2D gene because all patients were initially identified before 2013, before the broader genetic landscape of KS was fully understood. This limitation may impact our study's ability to fully capture the complete genetic heterogeneity of Kabuki syndrome, particularly concerning genes identified in later studies." (Page 6, Line 129-135)

  1. Results Section

The distribution of variants is well presented. Could the authors  explain why some variants are classified as "likely pathogenic" versus "pathogenic"? This would enhance the manuscript's accessibility, particularly for readers who may not be familiar with these classifications.

Ans: We have added an explanation of variant classification:

"Variants were classified as 'pathogenic' when they met strong criteria such as being null variants (nonsense, frameshift, canonical splice sites) or previously established as causative. 'Likely pathogenic' classifications were assigned to variants with strong evidence of pathogenicity but lacking definitive proof, such as novel missense variants at highly conserved residues with supporting functional studies. This classification was performed according to the American College of Medical Genetics and Genomics (ACMG) guidelines for the interpretation of sequence variants [14]." (Page 6, Line 137-143)

  1. Discussion Section

The manuscript suggests that environmental factors might influence phenotypic expression, this could be expanded with specific examples or hypotheses based on the literature. For instance, considering whether certain environmental exposures or lifestyle factors prevalent in Taiwan might contribute to the phenotypic differences observed between this cohort and others would provide a better analysis.

Ans: We have expanded on the potential environmental influences:

" Environmental factors that might influence KS phenotypic expression in Taiwan could include dietary habits, such as high consumption of iodine-rich seafood, which might impact thyroid function, a system often affected in KS. Additionally, the subtropical climate and associated pathogen exposure patterns could potentially influence the frequency and severity of recurrent infections observed in KS patients. However, these hypotheses require further investigation to establish any definitive links. Future studies focusing on the interaction between genetic variants and environmental factors in KS patients from different populations could provide valuable insights into the mechanisms underlying the phenotypic variability observed in this syndrome." (Page 9, Line 219-228)

We hope these revisions address your concerns and improve the quality and depth of our manuscript. Thank you again for your valuable input, which has undoubtedly enhanced our work.